Full Paper – MIDL 2026

# CSVR: Combined Surface and Volume Registration for Neonatal Brain MRI

**Saga N.B. Masui**[1,2] (iD)                                    SAGA.MASUI@KCL.AC.UK
**Yourong Guo**[1,2]                                              YOURONG.GUO@KCL.AC.UK
**Mohamed A. Suliman**[1]                                   MOHAMED.SULIMAN@KCL.AC.UK
**Mattias P. Heinrich** [3]                              MATTIAS.HEINRICH@UNI-LUEBECK.DE
**Nashira Baena**[2]                                         NASHIRA.BAENA@KCL.AC.UK
**Irina Grigorescu**[2]                                  IRINA.GRIGORESCU@KCL.AC.UK
**Logan Z. J. Williams** [1,2]                            LOGAN.WILLIAMS@KCL.AC.UK
**Ashleigh Davies**[1,6]                                   ASHLEIGH.DAVIES@KCL.AC.UK
**Vanessa Kyriakopoulou**[1,2]                     VANESSA.KYRIAKOPOULOU@KCL.AC.UK
**Gráinne McAlonan**[4]                                   GRAINNE.MCALONAN@KCL.AC.UK
**Jonathan O'Muircheartaigh**[2,4,5]                          JONATHANOM@KCL.AC.UK
**Emma C. Robinson**[1,2,5,6]                              EMMA.ROBINSON@KCL.AC.UK

[1] *Research Department of Biomedical Computing, School of Biomedical Engineering and Imaging Sciences, King's College London, London, SE1 7EH, UK*

[2] *Centre for the Developing Brain, School of Biomedical Engineering and Imaging Science, King's College London, London, SE1 7EH, UK*

[3] *Institute of Medical Informatics, Universität zu Lübeck, Germany*

[4] *Forensic and Neurodevelopmental Sciences, King's College London, London, SE5 8AF, UK*

[5] *MRC Centre for Neurodevelopmental Disorders, King's College London, London, SE1 1UL, UK*

[6] *Research Department of Early Life Imaging, School of Biomedical Engineering and Imaging Sciences, King's College London, London, SE1 7EH, UK*

**Editors:** Accepted for publication at MIDL 2026

## Abstract

Nonlinear image registration is a cornerstone of neuroimaging analysis, supporting both qualitative and quantitative comparisons of brain structures across individuals and over time. While traditional volumetric registration methods, driven by voxel intensities, achieve good alignment of subcortical regions, they generally fail to capture correspondences between highly convoluted and variable cortical shapes. Surface-based methods, which instead regularise mappings as geodesics along the cortical sheet, yield improved cortical alignment but ignore the subcortical domain, limiting their utility for whole-brain analyses. A unified registration framework would address these limitations to enable integrated analysis of cortical and subcortical structures and the neuronal fibres that connect them. However, achieving this is challenging, since matching heterogeneous cortical shapes implies large volumetric displacements local to the cortex. To overcome these challenges, we introduce CSVR, the first deep learning-based framework for combined surface–volume registration of neonatal MRI. By integrating hierarchical registration strategies with discrete optimisation, CSVR achieves accurate, smooth, and anatomically plausible alignment of the entire brain.

**Keywords:** Image registration, Surface-based cortical registration, Discrete optimisation, Volumetric registration, Neuroimaging,

## 1. Introduction

Accurate spatial alignment of medical images is essential for both cross-sectional comparison of anatomical structures and longitudinal tracking of morphological changes, making image registration a foundational component of medical imaging pipelines. Yet for neuroimaging, the heterogeneity of human cortical morphology contradicts the core principles of classical image registration frameworks (Glasser et al., 2016; Amunts et al., 2007), which assume that all images can be diffeomorphically mapped to a common coordinate space where analogous anatomical structures overlap. In reality, cortical folds vary across individuals in terms of their number, branching and orientations (Guo et al., 2025; Thompson et al., 1996; Ono et al., 1990; Guillon et al., 2024) and since this variability is often entirely natural and unrelated to disease or cognition, it translates into experimental noise that confounds downstream analysis.

One way of addressing this problem has been to selectively compare individual brains with others that share similar patterns of cortical folding (Meng et al., 2016; Duan et al., 2017). Taking this logic further, there has also been a move toward hierarchical registration frameworks (Guo et al., 2025; Ahmad et al., 2019a; Dong et al., 2018) that incrementally align brains of increasingly dissimilar shapes by e.g. clustering individuals into groups that share common anatomies; then using these groups to generate a *family* of templates that represent key modes of shape variation; before gradually registering subjects through this hierarchy until all brains are brought into a single reference space (Guo et al., 2025).

Since matching and aligning complex shapes in 3D is a highly ill-posed problem, this clustering approach can be made more tractable through the use of spherical registration (Suliman et al., 2022, 2025; Robinson et al., 2014, 2018; Yeo et al., 2009), which simplifies the problem of cortical alignment to one of matching feature maps defined over smooth 2D spheres (Fischl et al., 1999; Drury et al., 1996). In this way, spherical image registration methods have repeatedly demonstrated their capacity for improving cortical shape and/or cytoarchitectonic correspondence, relative to volumetric approaches (Glasser et al., 2016; Coalson et al., 2018), to support more precise characterisation of cortical asymmetries (Williams et al., 2023; Meyer et al., 2014; Raznahan et al., 2011), neurodevelopmental growth (Garcia et al., 2018; da Silva et al., 2025), and functional organisation (Glasser et al., 2016; Yeo et al., 2011; Kong et al., 2019).

However, while spherical mapping approaches may improve cortical alignment, they come at the cost of losing geometric correspondence to the rest of the brain, meaning that surface and volume analyses are typically run in disconnected template spaces that are not aligned and cannot be directly compared (Wu et al., 2018). These challenges disproportionately impact structural connectivity studies which seed from the cortex but project through white matter volumes.

A holistic framework for combined surface and volume mapping could address these limitations. Yet, achieving this is non-trivial due to the challenge of diffusing spherical warps—that allow sulci and gyri to overlap—into the volume without inducing extreme distortions. Previous solutions have attempted to regularise this task by either regularising volumetric warps to preserve spherical correspondences (Ahmad et al., 2019b), or by optimising very slow iterative processes that diffuse the warp into the volume (Postelnicu et al., 2008) with prohibitively long (∼17-hour) run-times. Other methods limit processing to within-subject

longitudinal alignment (Gibson et al., 2009) or alignment of twins (that share similar brain shapes)(Lepore et al., 2010).

One increasingly cited solution for non-convex image matching problems has been to use discrete optimisation, which reframes image matching as a combinatorial optimisation problem, in which the optimal displacement field must be selected from a quantised set of possible options. By constraining deformations in this way, these algorithms are able to reject locally appealing but globally inconsistent configurations, to model large and highly deformable transformations while maintaining anatomically plausible warps (Robinson et al., 2014, 2018; Besenczi et al., 2024; Glocker et al., 2008). Recent advances in deep discrete registration (e.g., DDR (Suliman et al., 2022); GeoMorph (Suliman et al., 2025); PDD (Heinrich, 2019)) take this one step further to combine the geometric robustness of such discrete formulations with the efficiency of deep feature extractors, making them particularly well-suited to coupled cortical and volumetric alignment.

**Contributions:** This paper presents a holistic framework for cluster-based deep-discrete surface and volumetric registration (CSVR) for precise anatomical alignment of individual human brains. It integrates deep discrete registration (DDR) (Suliman et al., 2022) of cortical surfaces with probabilistic dense displacement (PDD) (Heinrich, 2019) of MRI volumes within a single jointly-optimised framework to improve the alignment of whole-brain MRI more precisely than either surface or volume registration alone. We validate on neonatal data and show that the resulting warps are smooth and biologically plausible, and generalise robustly to unseen examples. In particular, CSVR performs well on all subjects, even more heterogeneous morphologies, where traditional volumetric methods show limited success and outperforms the classical adult baseline CVS.

## 2. Methods

Let $M = \{\mathbf{x}^M\}$ and $T = \{\mathbf{x}^T\}$ denote affinely aligned moving and target brain volumes, with $\mathbf{x}^M, \mathbf{x}^T \in \mathbb{Z}^3$ representing voxel coordinates. Let corresponding cortical surfaces be represented by mesh pairs $\partial M = (\mathcal{A}_M, \mathcal{S}_M^2)$ and $\partial T = (\mathcal{A}_T, \mathcal{S}_T^2)$ of corresponding anatomical and spherical surfaces that share vertex correspondences. In each case, anatomical surfaces $\mathcal{A}_M = \{\mathbf{v}_i^M \in \mathbb{R}^3\}_{i=1}^{N_v}$ and $\mathcal{A}_T = \{\mathbf{v}_i^T \in \mathbb{R}^3\}_{i=1}^{N_v}$ correspond to the inner cortical, or white matter, boundary defined from their corresponding volumes, from which spheres $\mathcal{S}_M^2 = \{\mathbf{s}_i^M \in \mathbb{R}^3\}_{i=1}^{N_v}$ and $\mathcal{S}_T^2 = \{\mathbf{s}_i^T \in \mathbb{R}^3\}_{i=1}^{N_v}$ are generated through learning-based spherical projection (Ma et al., 2025).

### 2.1. Deep Discrete Alignment

Deep discrete alignment frames image registration as a classification problem in which displacements—for a low resolution control point (CP) grid—are learnt through offering each (CP) $\{c_i\}_{i=1}^{N_c}$ a choice of target locations $\{l_i\}_{i=1}^{N_l}$ to deform to (Figure 2.1). For DDR, the control point grid and label points are defined from vertices derived from regularly tessellated icospheres. For PDD, the control point grid and label points are defined in 3D. Unlike classical discrete optimisation frameworks (Glocker et al., 2008; Robinson et al., 2014, 2018; Heinrich et al., 2015), the label space is made as wide as computationally possible such that the final deformation field may be classified in one shot. Regularisation is imposed through

deep conditional random fields (CRFs) that enforce spatial smoothness by encouraging neighbouring control points to deform to similar target locations (Zheng et al., 2015).

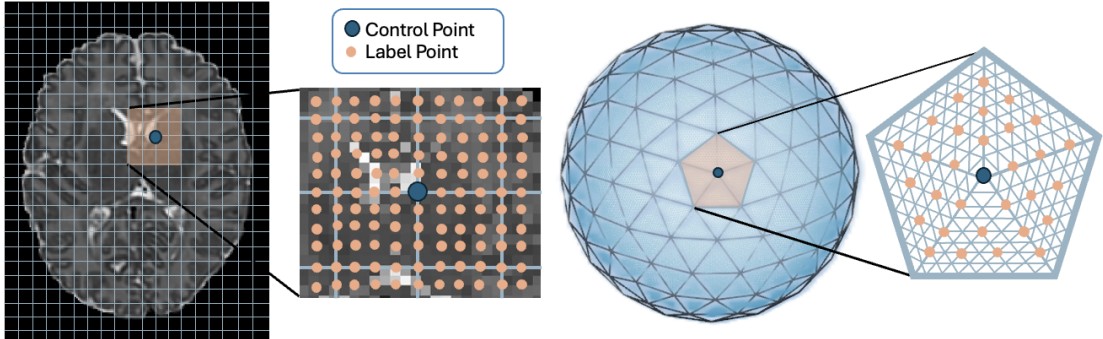

Figure 2.1: Control grids and label spaces of discrete registration in the volumetric (left) and spherical (right) domains.

## 2.2. Proposed Framework

Our overall goal is to estimate a mapping such that both internal intensities or MRI volumes and cortical surface features (and geometry) are aligned. CSVR (Figure 2.2) achieves this via:

### 2.2.1. SURFACE-DRIVEN ALIGNMENT (DDR)

Spherical alignment is performed using a pretrained DDR network from (Guo et al., 2025), that learns a spatial transformation $\phi_s : \mathcal{S}_M^2 \to \mathcal{S}_T^2$ that optimises overlap of sulcal depth functions $(f^M, f^T)$ over three stages: (1) a rotation network $h_R(f^M, f^T)$ first estimates a 3D rotation matrix $\mathbf{R}_S \in \mathbb{R}^{3 \times 3}$ that corrects the residual misalignment between spheres that remains after volumetric affine initialisation; (2) a non-rigid deformation module $h_{NR}(f^M \circ \mathbf{R}_S \mathcal{S}_M^2, f^T)$ is then trained to classify the target location of each control point $\{c_i^{DDR}\}_{i=1}^{N_c} \in \mathcal{S}_C^2$ by choosing from a fixed set of labels $\{l_i^{DDR}\}_{i=1}^{N_l} \in \mathcal{S}_L^2$, where the deformation grid $\mathcal{S}_C^2$ and the label grid $\mathcal{S}_L^2$ correspond to regularly tessellated icospheres of different resolutions $(|\mathcal{S}_L^2| > |\mathcal{S}_C^2|)$; (3) this process is smoothed through implementing CRF regularisation through a recurrent neural network (RNN) (Zheng et al., 2015), combined with a diffusion regulariser that operates on the spatial gradients. Both rotation and deformation modules employ spherical U-Net architectures (Ronneberger et al., 2015) that take sulcal depth features as input (resampled to a sixth-order isosurface) and learn MoNet convolutional filters (Monti et al., 2017). The final warp (output from $h_{NR}$) corresponds to a deformed spherical configuration $\mathcal{S}_M'^2$. To transfer this to the anatomical domain, we exploit vertex correspondence between $\mathcal{S}_M^2$ and $\mathcal{A}_M$ (and $\mathcal{S}_T^2$ and $\mathcal{A}_T$), and implement differentiable barycentric interpolation $\mathcal{I}_B$. Using the step-by-step framework from (Robinson et al., 2018) this allows moving spherical vertices $\mathbf{s}'^M_i$ to be located relative to target vertices $\mathbf{s}^t_i$, from which barycentric weights may be derived to project moving anatomical mesh vertices $\mathbf{v}_i^M$ onto the target anatomical mesh shape. This deformed surface $\mathcal{A}_M' = \{\mathbf{v}_i'^M \in \mathbb{R}^3\}_{i=1}^{N_v}$ is then used to extract a sparse displacement field $\{\mathbf{d}_i = \mathbf{v}_i'^M - \mathbf{v}_i^M\}_{i=1}^{N_v}$.

### 2.2.2. Surface-to-volume interpolation

From this sparse vertex-wise displacement we require a dense volumetric displacement field $\mathbf{u}_s(\mathbf{x})$, which is derived from a two-stage, differentiable Gaussian interpolation scheme. For each voxel location $\mathbf{x}^M$, we compute a weighted combination of displacements from nearby cortical vertices, such that:

$$\mathbf{u}_s^{\text{sparse}}(\mathbf{x}_j^M) = \sum_{i \in \mathcal{N}(\mathbf{x}^M)} w_{i,j}\mathbf{d}_i, \quad \text{with} \quad w_{i,j} = \exp\left(-\frac{\|\mathbf{v}_i^M - \mathbf{x}_j^M\|^2}{2\sigma_{\text{scatter}}^2}\right) \tag{1}$$

Here, $\mathcal{N}(\mathbf{x}^M)$ defines a cubic neighbourhood surrounding $\mathbf{x}^M$. This sparse field is then diffused into subcortical regions via a separable 3D Gaussian convolution with standard deviation $\sigma_{\text{vox}}$:

$$\mathbf{u}_s(\mathbf{x}) = G_{\sigma_{\text{vox}}} * \mathbf{u}_s^{\text{sparse}}(\mathbf{x}) \tag{2}$$

The final displacement field is normalised to preserve displacement magnitudes.

Finally, because volumetric registration networks typically predict *backward* deformation fields — i.e., mappings $\phi_v : T \to M$ that sample intensities as $M(\mathbf{x}^T + \mathbf{u}_v(\mathbf{x}^T))$, where voxels $\mathbf{x}^T \in T$ — this surface-derived field is first inverted before being used to transform $M$ and its corresponding segmentation onto the reference configuration, yielding preliminary warped volumes ($M'$) that propagate cortical correspondences between $A_M$ and $A'_M$ into the voxel domain.

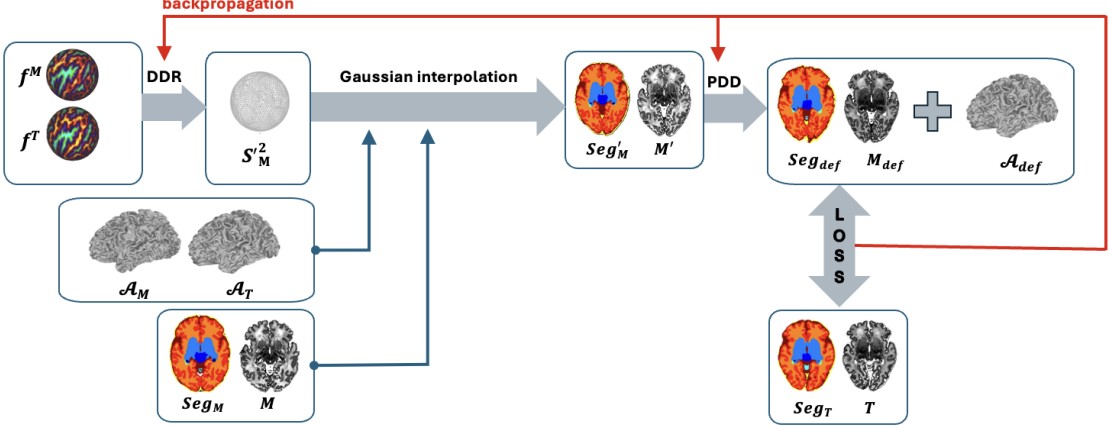

Figure 2.2: The CSVR framework: Moving and target sulcal depth maps are projected onto a regular icosphere and input to DDR to produce a deformed input sphere $\mathcal{S}'^2_M$. This spherical warp is used to project the moving anatomical mesh $\mathcal{A}_M$ onto the target anatomy $\mathcal{A}_T$ to return a deformed anatomy $\mathcal{A}'_M$ (with the shape of the target but topology of the input). Subtracting $A_M$ and $A'_M$ yields a sparse 3D displacement field $\{\mathbf{d}_i\}_{i=1}^{N_v}$, which is converted into a dense warp $\mathbf{u}_s$ through differentiable Gaussian interpolation, then inverted pull back intensities from the moving volume onto the target grid (as per convention) to generate a deformed and resampled input volume $M'$, which is used as inputs for PDD. This process is jointly optimised to learn a combined surface-and-volume transformation.

### 2.2.3. VOLUMETRIC SUBCORTICAL ALIGNMENT (PDD)

$M'$ is then passed to a pretrained probabilistic dense displacement network driven by image intensities $I^M$, $I^T$, and weakly supervised by tissue segmentation maps $Seg^M$ and $Seg^T$ (Heinrich, 2019). The model comprises three components: (1) an OBELISK-based feature extractor that produces intensity-invariant descriptors for $M'$ and $T$ (Heinrich et al., 2019), preceded by a 5×5×5 convolution that captures edge-like information; (2) a correlation layer that considers possible displacement vectors and evaluates feature dissimilarity from negated MSE across all candidates; and (3) architecturally embedded regularisation that alternates approximate min-convolutions and mean-field inference to simulate a Markov Random Field (Krähenbühl and Koltun, 2011; Zheng et al., 2015) and enforce smoothness. Ultimately, PDD uses the regularised MSE feature dissimilarities to predict a probability distribution over displacement labels at each control point via a softmax layer, which drives a non-local (MSE-based) segmentation loss. A continuous displacement field is then obtained through probability-weighted averaging and upsampling and subjected to a deterministic diffusion regularisation penalty. Since PDD was originally developed for abdominal CT registration, we adapt this method to subcortical brain MRI by complementing these terms with deterministic Dice and mutual information losses evaluated on the final warped image:

$$\mathcal{L}_{\text{total}} = \mathcal{L}_{\text{nonlocal}}(\text{Seg}_{\text{prob}}^{\text{LR}}) + \mathcal{L}_{\text{MI}}(\text{I}_{\text{det}}^{\text{LR}}) + \mathcal{L}_{\text{Dice}}(\text{Seg}_{\text{det}}^{\text{LR}}) + \mathcal{L}_{\text{reg\_diffusion}}(\text{I}_{\text{det}}^{\text{HR}}) \qquad (3)$$

Here, all losses are calculated at low resolution (LR), on either probabilistic or deterministic images and labels, but the diffusion regularisation is applied at high resolution (HR), and the approximated MRF implicitly regularises final deformation via the probabilistic weights of the candidate displacements. We also allow non-isotropic, dataset-adaptive control-grid dimensions and introduce a cortical deformation penalty to minimise interference with $\mathbf{u}_s(\mathbf{x})$. This penalty is defined as:

$$L_{cort} = \lambda_{cort} \sum_{\mathbf{x} \in \text{seg}(\mathbf{x}) == 2} \|\mathbf{u}_v(\mathbf{x})\|^2 \qquad (4)$$

Here, $\mathbf{u}_v(\mathbf{x})$ represents the volumetric deformation field, and $\lambda_{cort}$ is a tunable weight. This penalty is enforced only for moving locations indexed by cortical grey matter to limit displacements at locations where surface registration already provides accurate alignment.

### 2.2.4. JOINT OPTIMISATION AND WARP COMPOSITION

After pretraining DDR and PDD to separately optimise cortical and volumetric alignment, we then refine the models through differentiable Gaussian Interpolation (Section 2.2.2) into a unified model, CSVR, that jointly optimises both networks by backpropagating the PDD losses, complemented with an additional high-resolution Dice loss, through both networks.

## 3. Experimental Methods and Implementation
### 3.1. Data

This framework was trained on 681 subjects from the developing Human Connectome Project (dHCP) and independently validated on 100 subjects from the Brain Imaging in Babies study (BIBS) (Edwards et al., 2022). Out of these neonates, 648 were term-born (scan

age $41.5 \pm 1.7$ weeks, 297 female) and 133 were preterm-born (scan age: $41.3 \pm 1.7$ weeks, 59 female). Both datasets were acquired at the Evelina Newborn Imaging Centre using a Philips Achieva 3T scanner with a dedicated 32-channel neonatal head coil and positioning device (Glasser et al., 2013; Hughes et al., 2017). T2-weighted images were acquired in two stacks (sagittal/axial), with a repetition time (TR)/echo time (TE) of 12,000/156ms, flip angle of 90°, in-plane resolution of $0.8 \times 0.8$mm, slice thickness of 1.6mm and overlap of 0.8mm, and SENSE factor of 2.11 (axial) and 2.60 (sagittal). T1-weighted images were also acquired in sagittal and axial stacks with identical in-plane resolution and slice thickness, overlapped by 0.8mm (sagittal overlap of 0.74mm), and TR/TI/TE of 4,795/1,740/8.7ms, with SENSE factor of 2.27 (axial) and 2.66 (sagittal). These were motion-corrected and super-resolved to generate output volumes with 0.5mm isotropic grid sizes (Cordero-Grande et al., 2016, 2018). All babies were imaged during natural sleep.

### 3.2. Pre-processing

T2-weighted images were histogram-normalised and preprocessed using the dHCP Deep Learning–based Neonatal Pipeline (Ma et al., 2025), which builds on and is trained using outputs from the classical dHCP structural pipeline (Schuh et al., 2017; Makropoulos et al., 2018). This affinely aligns volumes to the dHCP 40-week neonatal T2w template (Schuh et al., 2018); then uses learning-based mesh fitting to shrink a template surface to the inner cortical (white) boundary, which is next reinflated to meet the outer cortical (pial) surface. White surfaces are subsequently inflated using a GPU-accelerated reimplementation of FreeSurfer's inflation algorithm (Fischl, 2012), from which sulcal depth maps are estimated by integrating over the functional. Finally, spheres are generated through spherical mapping, optimised using a spherical U-Net implementation similar to (Zhao et al., 2021). Prior to registration using CSVR, sulcal depth maps and white matter surfaces were resampled to ico-6 resolution, and all volumetric images were histogram normalised and resampled to $H = 176, W = 224, D = 160$.

### 3.3. Baseline (CVS)

The performance of the proposed CSVR framework was validated through comparison against CVS (Postelnicu et al., 2008). Since this requires all outputs in the FreeSurfer format, attempts to adapt our dHCP surfaces failed, requiring us to re-run surface generation with Infant Freesurfer (InfantFS) (Zöllei et al., 2020), which instead requires T1w images. For this, we first affinely registered T1w volumes to their corresponding template-aligned T2s, then performed InfantFS' recon-all processing, followed by CVS registration. The resulting CVS warpfields were applied to the T2w images and their DrawEM segmentations, enabling direct comparison with our CSVR method. Validation of CVS was performed using a smaller subset of 30 image pairs from the BIBS testing data (due to constraints of run time).

### 3.4. Clustering

To simplify the registration problem and minimise extreme distortions, subjects were clustered into groups of similar folding patterns using the method outlined in (Guo et al., 2025). In total, 90 clusters were used (30 for each of the frontal, parietal and temporal lobes) due to

observations that cortical folding variants do not co-occur, i.e. individuals that share similar folding patterns of the frontal lobes do not necessarily share variants of parietal or temporal lobes. Clusters were generated based on pairwise folding similarity, assessed through overlap of curvature following diffeomorphic alignment using DDR. Train and test datasets thus consisted of pairs of individuals with broadly similar folding patterns (for any of their lobes). Importantly, to avoid data leakage, all samples were separated into training (dHCP) and test (BIBS) subsets before pairing was performed, and each pair appeared twice—with samples used once as moving and once as target. Using the standard convention, the dataloader randomised the order in which pairs were presented during training.

### 3.5. Training and Model Implementation

**DDR (Surface Registration)**   The original DDR paper (Suliman et al., 2022) trained subject to template alignment. Therefore, here, we utilise the modified *pairwise* training scheme of (Guo et al., 2025). Data was input at high resolution - resampled to a sixth-order icosphere (ico-6) with 40,962 vertices. Labels were also defined at ico-6 resolution, and set to the 80 nearest neighbours of each control point, defined on an ico-2 grid (162 vertices). Following CRF optimisation, the deformed low-resolution control grid was upsampled to the input data resolution (ico-6) using barycentric interpolation. Optimisation was performed using Adam with a learning rate of $10^{-4}$, a diffusion regularisation weight of $\lambda = 1$, and performance was assessed via five-fold cross-validation on held-out subject pairs.

**PDD (Volumetric Registration)**   PDD was trained using: a diffusion regularisation weight $\lambda = 1$, dice weight of 0.5, MI weight of 1, a cortical penalty weight of $\lambda_{cort} = 0.001$, and non-local label weight of 15. The latter was scaled to account for inherent differences in magnitudes between the losses. We used control grids of size $44 \times 56 \times 40$ (which corresponds to sampling the input volume by 4). The network was optimised using the Adam optimiser with a learning rate of $5 \times 10^{-3}$, for 5 epochs over approximately 126k pairs. Model checkpoints were evaluated every 2000 iterations on a small validation subset containing at least one subject pair from each anatomical cluster, and the best-performing checkpoint was retained. Notably, we found that accurate alignment was achieved after only 2h 45 min of training, and more specifically, after the first 30k iterations. This aligns well with Heinrich (2019)'s original finding that PDD only requires a small number of subjects and a short training time.

**CSVR**   After independent pretraining, the combined model, CSVR, was trained on a subset of subjects representative of all anatomical clusters, but with permutations capped at 30 subjects per cluster, motivated by the promising generalisation performance of PDD. Training used the same hyperparameters as the pretrained networks, with the addition of a high-resolution Dice loss term (weight 0.5). $\sigma_{\text{scatter}}$ was set to 0.5 and $\sigma_{\text{vox}}$ to 12.0. These values were tuned manually by trading off extreme distortions (Jacobian determinants) against segmentation Dice scores on validation sets. Before producing a final output, the inverted surface warp was composed with the volumetric warp to produce a single deformation field $\mathbf{u}(\mathbf{x}) = \mathbf{u}_v \circ \mathbf{u}_s^{-1}$, applied to the original moving image as $M_{\text{registered}}(\mathbf{x}) = M(\mathbf{x} + \mathbf{u}(\mathbf{x}))$, and to the surface as $\mathcal{A}_{M,\text{registered}} = \{\mathbf{v}_i + \mathbf{u}^{-1}(\mathbf{v}_i)\}_{i=1}^{N_v}$. The purpose of this was to minimise interpolation artefacts.

## 4. Results and Discussion

We evaluate CSVR against five baseline methods: affine alignment (using ANTsPy (Avants et al., 2011)), FMRIB's Nonlinear Image Registration Tool, FNIRT (Andersson et al., 2007) and classical volume and surface alignment (CVS) (Postelnicu et al., 2008), with ablation performed by also comparing PDD and DDR (diffused into the volume). To keep comparisons as consistent as possible, FNIRT was configured with a control grid of the same resolution as PDD, and our whole-brain PDD network was trained using the parameters specified above. All learning methods were trained on pre-processed dHCP T2w MRI volumes and validated on the BIBS; FNIRT and ANTsPy were run on the BIBS test set only; whereas, CVS was necessarily run on T1w volumes (as described in section 3.1). The average inference time of CSVR was only 0.73s, compared to 32 min for FNIRT and 5.5 hours for CVS, which also required an additional 46 minutes for preprocessing with InfantFS.

**Dice overlap** of tissue labels following alignment are summarised in Table 1 and Figure 4.1 with statistical significance assessed using paired two-tailed t-tests, with false discovery rate (FDR) correction for multiple comparisons. Results show that CSVR achieved statistically significant improvements over all baseline methods across anatomical tissue types (all $p < 0.001$, FDR corrected) except deep grey matter compared to PDD ($p = 0.59$). These results demonstrate that our cortical surface constraints benefit not only cortical alignment but also propagate improvements to subcortical structures.

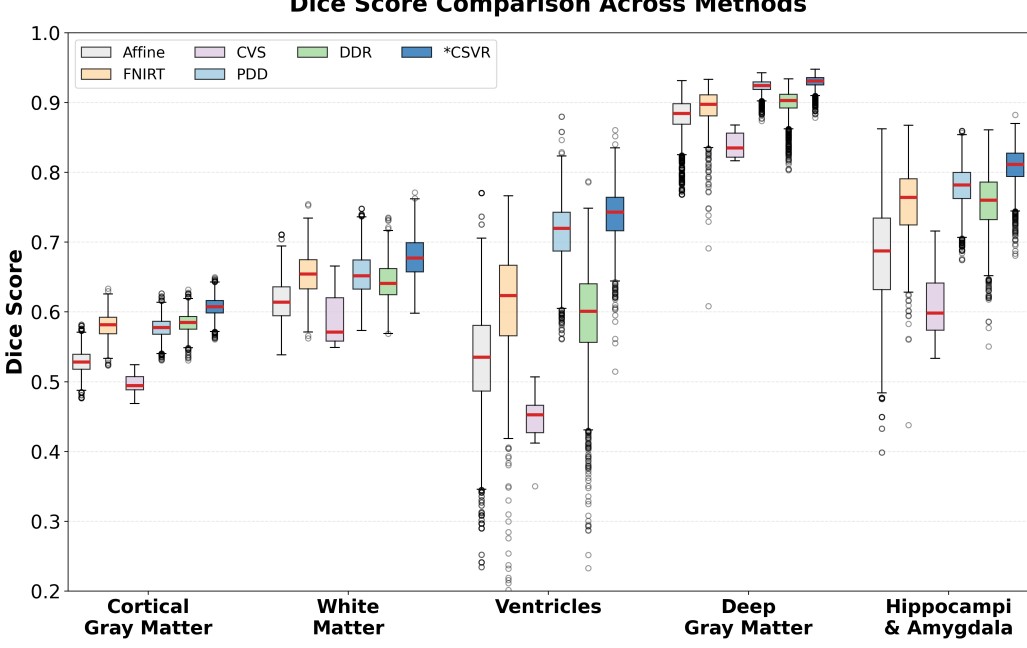

Figure 4.1: Regional Dice Overlap. On average, CSVR performs better than other methods across all tissue types, with the exception of deep grey matter, where alignment is on par with PDD. Notably, CVS performs poorly across all tissue types. For comparison, the average overall Dice across corresponding unregistered brains is 0.3659.

| Region | Affine | FNIRT | CVS | PDD | DDR | *CSVR |
|---|---|---|---|---|---|---|
| Cortical GM | $0.528_{\pm 0.041}$ | $0.580_{\pm 0.032}$ | $0.491_{\pm 0.021}$ | $0.577_{\pm 0.034}$ | $0.584_{\pm 0.022}$ | $\mathbf{0.615}_{\pm 0.020}$ |
| White Matter | $0.616_{\pm 0.052}$ | $0.653_{\pm 0.042}$ | $0.572_{\pm 0.040}$ | $0.654_{\pm 0.031}$ | $0.641_{\pm 0.042}$ | $\mathbf{0.661}_{\pm 0.031}$ |
| Ventricles | $0.529_{\pm 0.081}$ | $0.616_{\pm 0.102}$ | $0.452_{\pm 0.031}$ | $0.714_{\pm 0.063}$ | $0.727_{\pm 0.054}$ | $\mathbf{0.748}_{\pm 0.051}$ |
| Deep GM | $0.881_{\pm 0.031}$ | $0.890_{\pm 0.030}$ | $0.824_{\pm 0.041}$ | $0.923_{\pm 0.022}$ | $0.903_{\pm 0.021}$ | $\mathbf{0.924}_{\pm 0.012}$ |
| Hippocampi | $0.681_{\pm 0.071}$ | $0.753_{\pm 0.062}$ | $0.591_{\pm 0.052}$ | $0.780_{\pm 0.042}$ | $0.760_{\pm 0.031}$ | $\mathbf{0.803}_{\pm 0.032}$ |
| **Overall Mean** | $0.647_{\pm 0.051}$ | $0.698_{\pm 0.051}$ | $0.586_{\pm 0.041}$ | $0.730_{\pm 0.041}$ | $0.723_{\pm 0.031}$ | $\mathbf{0.750}_{\pm 0.031}$ |

Table 1: Dice Similarity Coefficient (DSC) across different brain regions (Mean ± Std).

**Distortions:** Meanwhile, results from Table A.2 demonstrate that CSVR achieves this with less than 0.15% negative Jacobians, relative to CVS with 1.4%. To investigate how this impacts alignment, Figures 4.2 and 4.3 show qualitative comparisons of image registration performance, highlighting that CVS' distortions often result in loss of cortical detail and misaligned subcortical structures, while CSVR maintains smooth and anatomically plausible alignments, with better overlap of segmentations and surfaces. We attribute CVS' failures to two factors: (1) the method was designed for adult rather than neonatal data, and (2) adaptation for use with InfantFS may have been suboptimal since this is primarily designed for neonates after birth (with only 5 training subjects aged under 2 months) (Zöllei et al., 2020). Despite selecting subjects with the best surface reconstructions, these results highlight that applying CVS to neonatal data is fundamentally limited by surface quality.

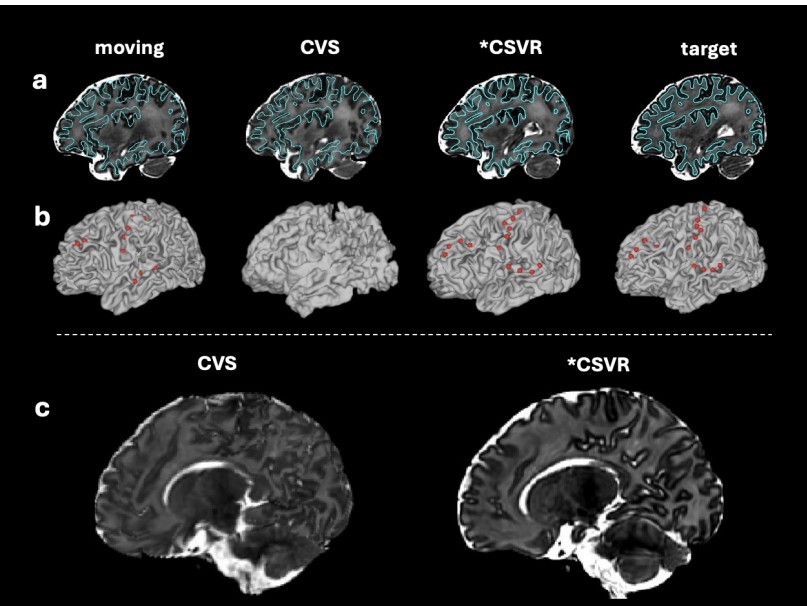

Figure 4.2: Visual comparison of CVS vs CSVR. (a) Target cortical surfaces overlaid on T2w images across registration stages. (b) Cortical surface reconstructions showing the target surface resampled onto moving topology, with red dots showing improved vertex correspondences after CSVR deformation; note the distortions on the CVS surface. (c) Sagittal views of deformed T2w images demonstrating CVS' anatomical distortions compared to CSVR.

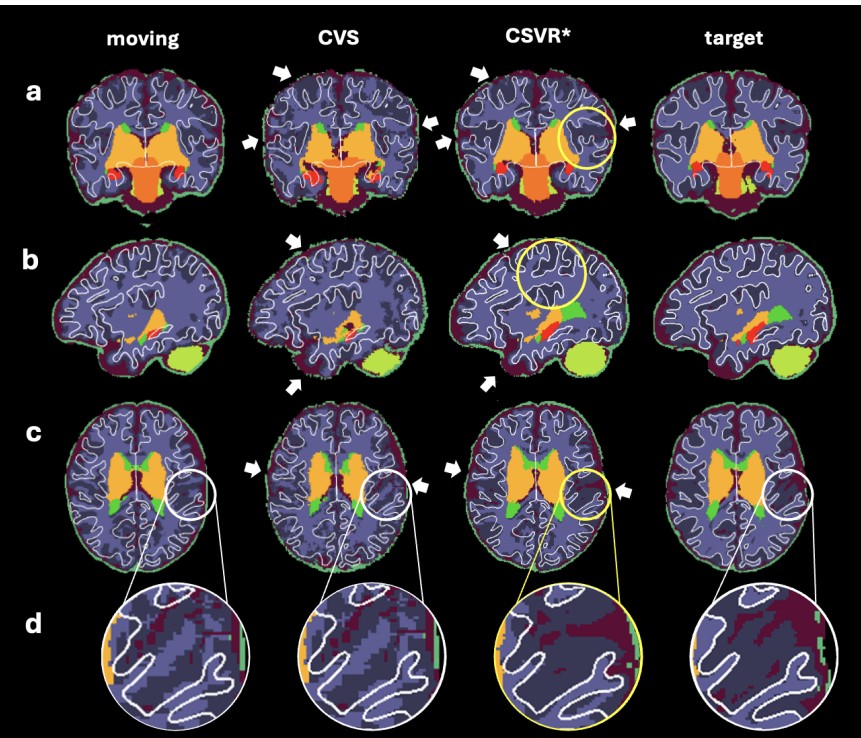

Figure 4.3: Qualitative comparison of CVS and CSVR registration across multiple views. Target cortical surfaces (white contours) overlaid on moving, CVS-deformed, CSVR-deformed, and target T2w images with segmentations in coronal (a), sagittal (b), and axial (c-d) orientations. White arrows and yellow circles highlight regions of improved alignment from CSVR, with magnified cortical details shown in (d).

**Improved gains for the most challenging subject pairs:** While subjects were grouped into registration clusters based on locally similar cortical folding patterns (at the lobar level), these do not always translate to similar whole-brain morphology. Consequently, some subject pairs exhibit substantially different overall brain shape, presenting challenging registration scenarios where affine or conventional non-linear methods struggle (Figure 4.4). Notably, CSVR maintains robust performance even for these difficult cases, with much improved alignment, in particular relative to affine and FNIRT, with up to 40% increase in dice overlap for some of the most challenging subjects. This pattern suggests that PDD's discrete optimisation framework confers robustness to local minima associated with highly heterogeneous brain morphology, while CSVR's cortical surface constraints provide additional gains primarily for cases where cortical folding patterns are informative for whole-brain alignment. Figure A.1 further highlights CSVR's ability to align difficult cases.

**Limitations and Future Work:** The current implementation of CSVR operates on relatively low-resolution control point- and label grids, which, while enabling rapid inference times, may limit the precision of fine-scale anatomical alignment. To achieve higher registration accuracy, future developments will include adopting a multi-stage optimisation strategy where surface and volumetric registrations are iteratively refined across progressively finer

resolutions. Such an approach would allow coarse alignment to establish global correspondence before fine-tuning local deformations, further improving Dice scores while maintaining computational efficiency.

**CSVR vs Other Methods: Per-Subject Comparison**

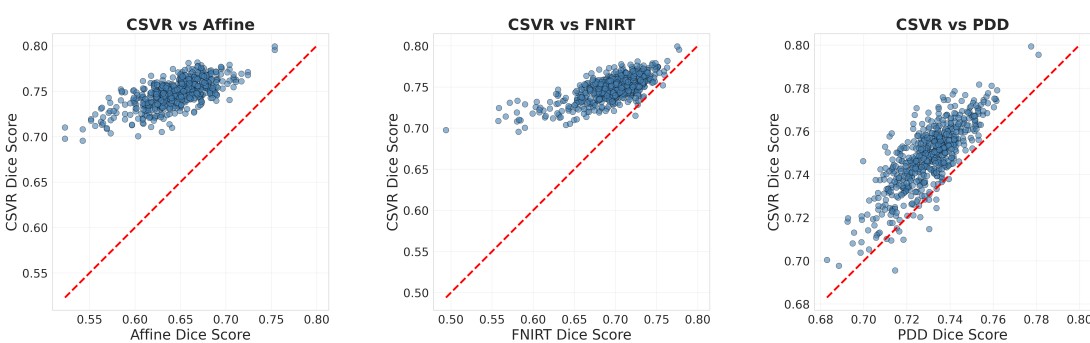

Figure 4.4: Per-Subject Comparison of CSVR vs other methods. Scatter plots comparing the average Dice score of CSVR against Affine, FNIRT, and PDD registration methods for individual subject pairs. The dashed red line represents the identity line $(y = x)$, where no performance change occurs. Points above the line indicate superior performance by CSVR. CSVR demonstrates widespread improvement over Affine and FNIRT across the entire range of registration difficulty, suggesting robust performance even in challenging cases. The performance difference is reduced when comparing CSVR to PDD for the most difficult subject pairs, indicating PDD's robustness to challenging whole-brain morphology.

## 5. Conclusion

In conclusion, we introduce CSVR, a deep discrete combined surface-volume registration network for brain MRI. Our method represents a significant step toward addressing the longstanding challenge of combined surface-volumetric template construction. Currently, whole-brain analyses are constrained by the lack of integrated templates, but by jointly optimising cortical surface and volumetric alignment, CSVR provides the foundational method necessary for generating such templates within Guo et al. (2025)'s hierarchical framework. While we have shown that CSVR is able to accurately register subjects within clusters of similar cortical folding, we have also shown that the model performs particularly well on more heterogeneous, difficult registration cases, showing great promise for downstream inter-cluster alignment and template creation.

Critically, CSVR outperforms the current combined registration baseline, CVS, which, designed for adult brains, struggles to align neonatal data. Moreover, CSVR achieves superior registration quality approximately 27,000 times faster than CVS, making it a practical solution for large-scale neonatal brain studies requiring combined surface-volumetric alignment. Finally, while we have validated CSVR on neonatal data, the main components of our method have demonstrated strong generalisability across diverse datasets: DDR and the MSM-HT hierarchical framework both show comparable performance on adult data, and PDD has already successfully generalised from abdominal CT to brain MRI, suggesting that CSVR should extend naturally to adult neuroimaging applications.

## Acknowledgments

The authors would like to acknowledge the participants of the dHCP and BIBS studies. The dHCP neonatal dataset was provided by the developing Human Connectome Project, KCL-Imperial-Oxford Consortium, funded by the European Research Council (ERC) under the European Union Seventh Framework Programme (FP/2007-2013) / ERC Grant Agreement no. [319456]. The BIBS data were funded by EU-AIMS (European Autism Interventions)/EU AIMS-2-TRIALS, an Innovative Medicines Initiative Joint Undertaking under Grant Agreement No. 777394. The School of Biomedical Engineering and Imaging Sciences is supported by the Wellcome EPSRC Centre for Medical Engineering at King's College London (WT 203148/Z/16/Z) and the Department of Health via the National Institute for Health Research (NIHR) comprehensive Biomedical Research Centre (BRC) award to Guy's & St Thomas' NHS Foundation Trust in partnership with King's College London and King's College Hospital NHS Foundation Trust. We acknowledge infrastructure support from the NIHR Mental Health BRC at South London and Maudsley NHS Foundation Trust, King's College London.

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

## Appendix A.

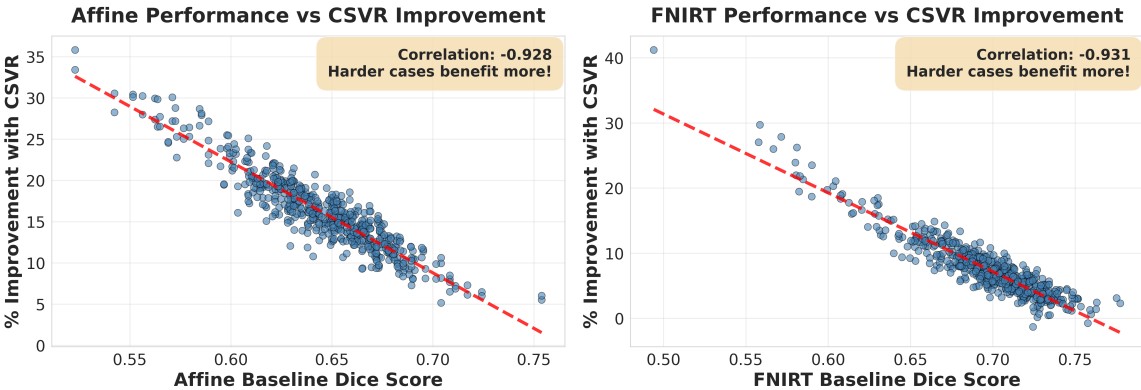

Figure A.1: Performance of other models (Affine and FNIRT) vs % Improvement with CSVR. Harder cases for which FNIRT and affine alignment is poor, benefit more from CSVR registration.

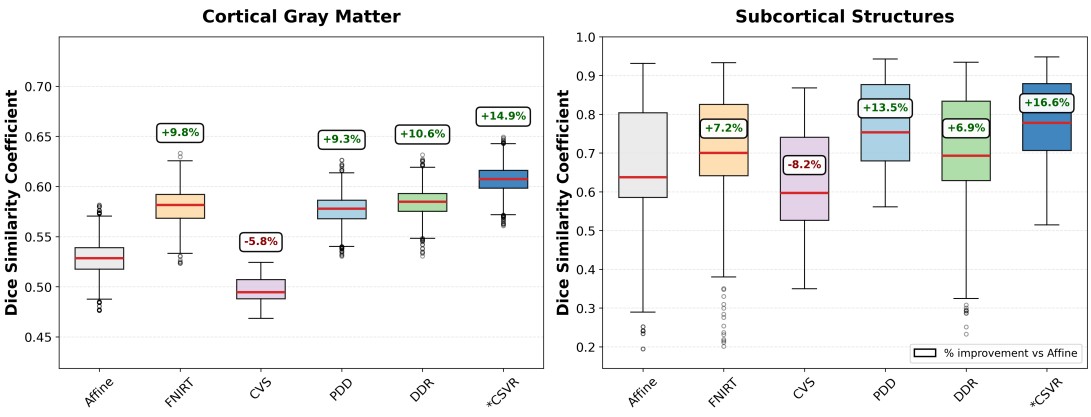

Figure A.2: Dice scores from Figure Figure 4.1 summarised into cortical vs subcortical box plots. Green and red percentages show mean improvement compared to the affine baseline.

| Method | Neg. Jac. (%) | Min Jac. | Mean Jac. | Std Jac. |
|--------|--------------|----------|-----------|----------|
| Affine | 0.000 | 1.000 | 1.000 | 0.000 |
| FNIRT | 1.854 | $-4.167$ | 0.996 | 0.634 |
| CVS | 1.401 | $-102.661$ | 0.473 | 6.186 |
| PDD | 0.191 | $-1.496$ | 0.994 | 0.345 |
| DDR | 0.001 | $-0.110$ | 0.983 | 0.128 |
| *CSVR | 0.141 | $-1.253$ | 0.997 | 0.346 |

Table A.2: Jacobian Determinant Statistics by Method

