# OpenReview forum: "CSVR: Combined Surface and Volume Registration for Neonatal Brain MRI"
_MIDL.io/2026/Conference — MIDL 2026 Poster_

### Official Review · Reviewer_mAGF · 2026-01-07

**Confidence:** 4
**Preliminary Rating:** 3
**Final Rating:** 4

**Summary:**

The paper proposes a deep learning method for non-linear surface-volume registration integrating hierarchical registration techniques with discrete optimization. More specifically, it presents a framework for cluster-based deep discrete surface for cortical surfaces and probabilistic dense displacement for volumetric registration. The paper presents quantitative results on the dHCP dataset.

**Strengths:**

1. Proposes a novel deep learning method (joint optimization) for combining surface and volumetric registration.
2. Combines 2 well-studied concepts in a framework and performs registration on neonatal brains.
3. It is comprehensible, pleasant to read, and provided link to GitHub.
4. Has a nice presentation of quantitative results.

**Weaknesses:**

1. I find the paper title slightly misleading talking about brain MRI but only presenting results in neonatal brains. I would much more prefer to write neonates in the title or include an adult brain dataset.
2. The paragraph before contributions seems to highlight the literature review. However it is not very clear what the limitations of the current literature is and how all these methods differ from the proposed one. I would like to invite the authors to clarify this.
3. Although the paper impressively claims that the inference takes 0.73s and the negative Jacobeans are less than 0.5% it would be nice to see how this is differrent from the compared methods.
4. Why did the authors choose as a baseline the older Suliman et al. work instead of the more recent?
5. Although I quite like the way the quantitative results are presented I believe folding ration and inference time would be nice to be reported in Table 1 along with the std for DCS. Moreover in figure 4.1 It would be nice to see the mean for surface, volumetric, overall along with the reported means.
6. I would like to invite the authors to also provide some qualitative results that support their claims. Since registration does not have ground truth qualitative assessment along with accuracy and regularity is needed to determine how good the result is.
7. Not sure if I am getting it wrong but I believe the most common and accurate way so far to do combination of surface and volume registration is the CVS method of Postelnicu et al. I understand that it might take long to run experiment however I believe that this baseline is really missing from this paper.

**Detailed Comments:**

Please see the weaknesses above.

**Justification Of Final Rating:**

I would like to thank the authors for their responses and the effort they made to change their paper. I believe that the manuscript has been substantially improved. I would therefore like to raise my score to weak accept.

**Justification Of The Preliminary Rating:**

The paper presents a conceptually interesting approach, introducing a new way of jointly training surface and volume registration. It has a clear motivation and addresses a gap in the literature.

I really enjoyed reading the paper, and I found it interesting and novel. However, I would like to draw the author’s attention to a concurrent work by Abulnaga et al (Unified Brain Surface and Volume Registration) https://arxiv.org/pdf/2512.19928v1 which might be of interest to then. Moreover, I tend towards borderline, especially because I believe the CVS baseline is missing, but I am willing to improve my score if the authors try to incorporate some of the suggestions above.

**Questions To Address In The Rebuttal:**

Please see the weaknesses above.

---

> ### Author Response · Authors · 2026-01-24
> **We thank the reviewer for their comments on novelty and readability. We address specific points and queries below.**
>
> ## Weakness 1 (W1): Title Specificity
>
> We agree the title should reflect our current focus on neonatal alignment. We will revise to: "CSVR: Combined Surface and Volume Registration for Neonatal Brain MRI".
>
> ## W2: Literature Review Clarity
>
> We thank the reviewer for this comment. To clarify, we are not setting out to state the limitations of previous methods but rather explain why this is a necessary yet challenging problem to solve and provide a history of progress on this problem to date, while motivating our choice of learning framework. With respect to the limitations of previous methods, from running the requested benchmarking experiments against CVS, we can state that this does not work out-of-the-box for neonates and would require adaptation, and regardless, this is a multi-step process that takes many hours to run. Similarly, to other classical methods that have tackled this problem, such as Ahmad et al. (2019), these methods frame the problem as one of constraining a volumetric warp with the output of a spherical warp rather than performing joint optimisation.
>
> ## W3: Jacobian Comparison Across Methods
>
> We agree that a more thorough Jacobian analysis was lacking and apologise for this oversight. We have added detailed Jacobian statistics for CSVR and all baselines—in the results paragraph "Distortions:"
>
> ## W4: Choice of DDR over GeoMorph
>
> We chose DDR over GeoMorph because it has been previously used to train the pairwise alignment for the hierarchical registration framework that CSVR integrates with (Guo et al. 2025), which benchmarks on our dataset. It may be worth emphasising that GeoMorph's primary advantage is for multimodal registration through learned feature weighting—for unimodal sulcal depth alignment (our use case), with limited benefit over DDR for shape alignment.
>
> ## W5: Reporting Format (Table 1 & Figure 4.1)
>
> We have added: (1) standard deviations for all Dice scores in Table 1, (2) inference times for all methods and (3) separate reporting of surface-based and volumetric overall mean metrics in Figure A2.
>
> ## W6: Qualitative Results
>
> As per your suggestion, we have added qualitative image comparisons showing (1) deformed images and their corresponding segmentations with target surfaces overlaid, (2) closeup comparisons of CVS vs CSVR's warped T2s (3) vertex correspondences across moving, target and deformed surfaces.
>
> ## W7: CVS Baseline (Postelnicu et al.)
>
> We have added a comparison against FreeSurfer's CVS to validate our methodological contribution against the established baseline. We find it fails on these neonatal cases, due to a culmination of reasons: firstly, we were unable to translate dHCP outputs into a form compatible with CVS; secondly, although we were able to run infant FreeSurfer for those datasets for which T1s were available, the resulting surfaces were not as good as those available from the bespoke neonatal dHCP pipeline; third it seems likely that the default parameterisation of CVS was not designed for neonatal datasets.

---

> > ### Comment · Reviewer_mAGF · 2026-01-27
> >
> > I would like to thank the authors for their responses and efforts. I believe that the manuscript has been substantially improved. I would therefore like to raise my score to weak accept.

---

### Official Review · Reviewer_XSxB · 2026-01-11

**Confidence:** 4
**Preliminary Rating:** 5
**Final Rating:** 5

**Summary:**

This paper proposes a registration framework that enables the integrated analysis of cortical and subcortical brain structures. The method combines deep discrete registration (DDR) for cortical surface registration with probabilistic dense displacement (PDD) for subcortical volumetric MRI registration, jointly optimized within a unified framework. The approach is trained and evaluated on relatively large datasets, comprising 681 subjects from the dHCP dataset and 100 subjects from the BIBS dataset. Experimental results show improvements of the proposed CSVR method over the baseline approaches.

**Strengths:**

1) The paper is clear, well-written, and easy to follow. The introduction and methods sections provide sufficient coverage of the relevant literature, making the work accessible to readers from diverse backgrounds. The research motivation on combining cortical and subcortical brain MRI registration is relevant and well-justified.

2) The proposed methodology is technically sound. The use of discrete optimization to jointly optimize surface and volumetric registration is a reasonable design choice, as it allows the optimizer to consider all candidate label points and decide for itself whether to prioritize deforms in the subcortical region or the cortical region to avoid conflicts.

3) The training and evaluation data are relatively large, which is commendable for the scope of a conference paper and significantly solidifies the experimental results.

**Weaknesses:**

Some minor comments:
1) The paper primarily relies on existing and widely adopted techniques from the literature, such as DDR and PDD, which somewhat limits the novelty of the proposed framework.

2) The data curation, preprocessing, and clustering pipeline involves multiple stages. Consequently, although the reported inference time of the CSVR framework is impressive (under one second), the overall end-to-end runtime may be substantially increased by the complexity of the preprocessing phase. Moreover, failures or inaccuracies during preprocessing could propagate and adversely affect the subsequent training and inference stages. Further discussion or analysis of this aspect would help strengthen the practical applicability of the proposed method.

3) The authors choose discrete optimization as the main “glue” for joint surface and volume registration, seemingly by default, as it is commonly preferred for solving non-convex image matching problems. Providing a more explicit justification for this choice, particularly through comparison with alternative strategies such as continuous or hybrid optimization, would strengthen the methodological rationale of the proposed framework.

4) The authors mentioned that combined surface and volume mapping would introduce extreme distortions. To better substantiate the benefits of the proposed joint optimization strategy, it would be valuable to include an experiment comparing the proposed approach with a baseline that simply composes independently pretrained DDR and PDD deformation fields.

**Detailed Comments:**

No further comment.

**Justification Of Final Rating:**

The authors have sufficiently addressed my minor concerns in both the revised manuscript and the comment section. I am willing to keep my final rating unchanged from my initial assessment of “strong accept”.

**Justification Of The Preliminary Rating:**

The proposed methodology is technically sound. The use of discrete optimization to jointly optimize surface and volumetric registration is a reasonable design choice, as it allows the optimizer to consider all candidate label points and decide for itself whether to prioritize deforms in the subcortical region or the cortical region to avoid conflicts. The training and evaluation data are relatively large, which is commendable for the scope of a conference paper and significantly solidifies the experimental results. Experimental results show improvements of the proposed CSVR method over the baseline approaches in both accuracy and runtime.

**Questions To Address In The Rebuttal:**

See the Weaknesses section above.

---

> ### Author Response · Authors · 2026-01-24
> **We thank the reviewer for their support and address residual concerns below.**
>
> ## W1: Novelty (Reliance on DDR and PDD)
>
> While CSVR does build from pre-existing architectures (DDR and PDD), composing these within a joint alignment framework for neonatal brains was non-trivial. The first challenge was to adapt PDD, which has previously only been used for abdominal CT to brain alignment. Subsequently, joint optimisation required all composition components to be made differentiable: this includes Gaussian and Barycentric interpolation, warp inversion and composition.
>
> ## W2: Preprocessing Complexity and Runtime
>
> The reviewer correctly notes that preprocessing does add to end-to-end runtime. We clarify: (1) surface extraction uses the dHCP deep learning pipeline (~2 min/subject), (2) clustering is performed once per dataset as an offline step.
>
> Importantly, these runtimes are short compared to classical alternatives—FreeSurfer's CVS requires ~17 hours per registration for adults and ~5.5 hours for neonates (excluding preprocessing which adds an extra 46 minutes, on average). As such, the 0.73s CSVR inference time (given preprocessed inputs) represents a substantial practical advantage. We also note that preprocessing errors (e.g., failed segmentation) affect volumetric-only methods similarly, as they generally stem from the same upstream challenges (motion, partial volume effects).
>
> ## W3: Justification for Discrete optimisation
>
> Despite the well-established advantages of discrete optimisation for non-convex matching problems, continuous methods like VoxelMorph remain the default in deep learning registration. We chose discrete optimisation specifically because its robustness to local minima makes it highly suitable for the challenging task of aligning heterogeneous cortical morphology, where folding patterns vary substantially across individuals. This robustness is particularly critical when combining surface and volumetric registration, as the composed deformation must handle large cortical displacements while maintaining anatomical plausibility.
>
> ## W4: Joint optimisation vs. Independent Composition
>
> We have added a DDR-only ablation that applies surface-derived warps to the volume without PDD refinement, demonstrating clear advantages for joint optimisation. For this rebuttal period, we prioritised benchmarking against the classical baseline (CVS), but we agree with the reviewer that an experiment comparing the proposed approach with a baseline that simply composes independently pretrained DDR and PDD deformation fields would be useful for future work, especially after implementing iterative registration and making our interpolation parameters learnable and spatially-varying to support more flexible alignment.

---

> > ### Comment · Area_Chair_WtS9 · 2026-02-01
> > **Update the final ratings**
> >
> > Please kindly review the authors’ rebuttal and update the final rating by February 1, 2026 (23:59 AoE).

---

### Official Review · Reviewer_6fe3 · 2026-01-12

**Confidence:** 2
**Preliminary Rating:** 4
**Final Rating:** 5

**Summary:**

The paper proposes CSVR, a deep learning registration approach that combines cortical surface registration with volumetric registration for neonatal MRI. A surface warp is converted into a dense 3D deformation field and then jointly fine-tune with a probabilistic volumetric model, yielding a composed deformation. Experiments on neonatal datasets show better tissue overlap (Dice) than standard affine/FNIRT and also improvements over the volumetric deep baseline. Inference is fast and deformations mostly well-behaved (low negative Jacobian rate). Overall, the key significance is a single consistent alignment for cortex + subcortex, which is useful for downstream analyses that mix both

**Strengths:**

- Good idea / right problem: combining surface and volume registration is well motivated for neonatal brains, where cortex is tricky for volume methods.

- Solid empirical evidence: Dice improves over common baselines and the deep volumetric model, suggesting the coupling is actually doing something.

- Practical: fast inference and plausible warps makes this very usable.

**Weaknesses:**

- Missing ablations: hard to tell which component matters most (joint fine-tuning? cortical penalty? surface to volume interpolation?) without a couple of simple removals.
- cope / generalisation unclear: evaluation is neonatal-specific and seems to rely on within-cluster pairing; it’s unclear how robust this is to more heterogeneous anatomy
- Inverse-warp and regularity details: the use of $u_s^{-1}$ is central, but how the inverse is computed and when it might fail is not fully clear to me. Also low negative Jacobians is great but not the same as a diffeomorphic guarantee, some discussion of failure modes would help.

**Detailed Comments:**

- Please specify the Gaussian interpolation hyperparameters, were they tuned or fixed across datasets?
- Clarify how the cortical deformation penalty is defined and how strong it is relative to other losses (weighting), since this seems central to “don’t destroy the surface-driven correspondences”
- Dice is fine, but for registration it could be helpful to add at least one more measure: e.g. surface metrics (sulcal depth correlation or curvature alignment), boundary distance, or a Jacobian-based smoothness summary beyond “negative %”
- Minor but important: there are several typos/inconsistencies (e.g., “CVSR” vs “CSVR”, “regsitra-tion”, “acknowlege”). Please fix for camera-ready; it distracts from an otherwise solid paper

**Justification Of Final Rating:**

The authors have addressed my comments adequately. In my opinion they have also addressed most of the comments of the other reviewers (as far as I can say). Taken together, this is making the contribution considerably stronger.

**Justification Of The Preliminary Rating:**

I like the paper: it tackles a real bottleneck (cortex vs whole-brain alignment) with a coherent combined approach and shows consistent improvements over strong baselines, while being fast enough to matter in practice. What can be improved to make it a strong accept: are missing ablations that would make the contribution crisper, and limited evidence about robustness/generalisation beyond the specific neonatal and clustering regime.

**Questions To Address In The Rebuttal:**

- Ablation / attribution: What is the Dice gain vs PDD if you (a) remove joint fine-tuning, (b) remove the cortical penalty in PDD, (c) change or remove the surface-to-volume interpolation, or (d) apply DDR-only warp to the volume as a baseline? Even 1–2 targeted ablations on a subset could help a lot.

- Inverse warp details: How exactly is $u_s^{-1}$ computed in practice, and how stable is it? Any cases where inversion fails or introduces folding (negative Jacobians)?

- Generalisation beyond clusters: Do you have any evidence (even small-scale) that CSVR helps for inter-cluster alignment or more heterogeneous anatomy than the within-cluster regime?

- Surface alignment metrics: Since a core claim is improved cortical correspondence, can you report at least one surface-based alignment measure (not only volumetric tissue Dice)?

---

> ### Author Response · Authors · 2026-01-24
> **We thank the reviewer for their positive assessment and recognition that CSVR tackles a real bottleneck for clinical neonatal brain imaging research with what they consider a coherent approach. We address each question below.**
>
> ## Q1: Ablation Studies
>
> Following the reviewer's suggestion, we have included DDR-only ablation (by propagating the surface warp into volume without PDD refinement or joint optimisation). This isolates the surface registration contribution and demonstrates each component's role. Due to time constraints for this revision, additional ablations are noted as future work.
>
> ## Q2: Inverse Warp ($u_s^{-1}$) Details
>
> We clarify that no explicit numerical inversion is computed. The DDR output warp is diffeomorphic and interchangeable by construction, so we simply use barycentric interpolation to resample the moving mesh topology onto the target anatomical surface (shape); then compute displacement as $d_i = v_i^{\\mathrm{moving}} - v_i^{\\mathrm{resampled\\\\_target}}$ directly yielding the backward field for image resampling (but the inverse of what we would use to deform the moving surface vertices to the target). Jacobian analysis (now included for all methods, including DDR-only) confirms that this step only results in 0.001% negative Jacobians (see Table A2).
>
> ## Q3: Generalisation Beyond Clusters
>
> CSVR is designed for within-cluster registration as part of a hierarchical framework. Guo et al. (2025) demonstrated that hierarchical inter-cluster alignment produces sharper group averages and improved correspondence surface-based analyses, and similar findings have been reported for volumetric alignment in Dong et al. (2018), for example. However, that being said, it's worth noting that clusters are defined based on similar folding patterns within each of the frontal, temporal and parietal lobes, without any guarantee of co-occurrence, meaning that just because two individuals share similar folding patterns in their frontal lobes, it does not necessarily mean they share similar patterns in their temporal or parietal lobes. As such, much of our alignment still involves deforming heterogeneous anatomies, and we show that CSVR, most likely by leveraging discrete optimisation, performs very well compared to baseline methods for these more difficult cases.
>
> ## Q4: Surface Alignment Metrics
>
> We have added images of our deformed anatomical surfaces to our qualitative results, showcasing improved vertex correspondences following registration (see Figure 4.2b).
>
> ## Typos
>
> Thank you for pointing out the typos. We have fixed: CVSR→CSVR, regsitration→registration, acknowlegee→acknowledge.

---

> > ### Comment · Reviewer_6fe3 · 2026-01-31
> >
> > Thanks to the authors for their comments and for the revision. I think my comments have been addressed. I would therefore (also in light of the other reviews and responses) increase my score.

---

### Official Review · Reviewer_bLqu · 2026-01-14

**Confidence:** 3
**Preliminary Rating:** 3
**Final Rating:** 4

**Summary:**

This paper introduces CSVR, a deep learning framework that jointly optimizes cortical surface registration (via Deep Discrete Registration, DDR) and volumetric registration (via Probabilistic Dense Displacement, PDD) for whole-brain MRI alignment. The key technical contribution is a differentiable pipeline that converts sparse surface displacements into dense volumetric fields through Gaussian interpolation, followed by 3D convolution. The method is evaluated on neonatal brain MRI from dHCP (n=681) and BIBS (n=100) datasets, achieving mean Dice improvements of +0.038 over standalone PDD for cortical gray matter. The authors demonstrate 0.73s inference time and claim robust performance on morphologically challenging cases, positioning this as foundational work toward integrated surface-volumetric brain templates.

**Strengths:**

1. **Clinically relevant problem**: The disconnect between surface-based cortical registration and volumetric subcortical registration is a genuine limitation affecting structural connectivity studies. The motivation is well-articulated with appropriate contextualization.

2. **End-to-end differentiable architecture**: Making all operations differentiable (barycentric interpolation, Gaussian interpolation, warp inversion, composition) enables joint optimization where PDD losses influence DDR feature learning, which is elegant.

3. **Computational efficiency**: The 0.73s inference time represents a major practical advantage over iterative methods, enabling potential clinical adoption.

4. **Robustness analysis**: Figures 4.2 and A.1 convincingly demonstrate that CSVR provides the largest improvements where baseline methods struggle most, suggesting targeted benefit for difficult cases.

5. **Principled use of hierarchical clustering**: Following Guo et al. (2025), the recognition that registering highly dissimilar brains is ill-posed demonstrates methodological sophistication.

**Weaknesses:**

1. **Missing methodological details**: A cortical deformation penalty is mentioned but never mathematically defined. The interpolation parameters $\sigma_{\text{scatter}}$ and $\sigma_{\text{vox}}$ (Equations 1-2), which likely significantly affect performance, are not specified or ablated. The warp inversion procedure is not described despite being non-trivial.

2. **Lack of standardized baselines**: The submission would benefit from comparing to established volumetric image registration baselines, such as VoxelMorph, DIRNet, or LapIRN. DDR-only results are also absent, preventing isolation of each component's contribution.

3. **Topologically problematic interpolation**: Gaussian interpolation in 3D Euclidean space ignores cortical topology. Points close in 3D (e.g., opposing sulcal banks) may have very different displacements—the kernel would erroneously blend contradictory vectors for intervening voxels. Did the authors consider the option of geodesic distance-based blending?

4. **Narrow evaluation scope**: Dice is the only evaluated metric, while also being a training objective. Additional metrics such as ASSD, a Jacobian analysis (claimed but not shown), and sulcal depth correlation would strengthen the results.

**Detailed Comments:**

- Typo: "regsitration" in Section 5; "CVSR" vs "CSVR" inconsistency in Figure 2.2 caption
- The claim of "statistically significant improvements...across anatomical tissue types (all p < 0.001)" contradicts the stated p=0.59 for deep gray matter
- Figure 4.2 shows cases where CSVR underperforms PDD—characterizing these failures would strengthen the paper
- The 90-cluster scheme (30 per lobe) lacks justification; sensitivity analysis would be valuable
- Consider discussing how surface extraction errors propagate through the pipeline
- The paper would benefit from explicit acknowledgment that CSVR currently solves a constrained (within-cluster) registration problem rather than general pairwise registration

**Justification Of Final Rating:**

I appreciate the efforts of the authors to address my concerns, especially w.r.t. additional baselines and ablations, which are fair and appropriately answered. As a response, I will gladly update my score from borderline to weak accept.

**Justification Of The Preliminary Rating:**

The paper addresses an important and well-motivated problem with a technically reasonable approach. However, the evaluation lacks modern deep learning baselines (VoxelMorph, LapIRN), uses only a single metric on a narrow data domain, and is missing critical methodological details needed for reproduction. The improvements over standalone PDD are modest (+0.038 Dice for cortical GM) and not significant for deep structures (p=0.59). While the computational efficiency and architectural design are genuine strengths, the gap between claims and evidence is relatively large. With experiments on modern baselines, additional evaluation metrics, and a complete methodological specification, this could become a solid contribution.

**Questions To Address In The Rebuttal:**

1. **What is the mathematical formulation of the cortical deformation penalty?** Please specify the functional form and weight used.

2. **What are the values of $\sigma_{\text{scatter}}$ and $\sigma_{\text{vox}}$?** How sensitive is performance to these choices?

3. **Could additional volumetric image registration baselines be provided** given that e.g. VoxelMorph is readily available and should be quick to train?

4. **Can you show DDR-only results** to isolate the contribution of surface vs. volumetric components?

5. **Can you provide the Jacobian analysis** (negative Jacobian percentages, determinant statistics) that is claimed but not shown?

---

> ### Author Response · Authors · 2026-01-24
> **We thank the reviewer for their detailed review of this paper and for highlighting the key strengths of our work. We appreciate their constructive feedback and have addressed the key points below.**
>
> ## Q1&2: Missing methodological details and typos
>
> We apologise for the oversight. Our cortical deformation penalty is defined as:
>
> $$L_{\\text{cort}} = \\lambda_{\\text{cort}} \\sum_{x \\in \\text{seg}(x)==2} ||u(x)||^2$$
>
> where $u(x)$ represents the volumetric deformation field and this penalty is only enforced for locations indexed by cortical grey matter (of the moving image) to penalise displacements at locations where surface registration already provides accurate alignment, and $\\lambda_{\\text{cort}}=0.001$.
>
> We interpolate the surface warp into the volume using $\\sigma_{\\text{scatter}}=0.5$ and subsequently diffuse this sparse field towards subcortical regions using $\\sigma_{\\text{vox}}=12.0$. These values were tuned manually through trading off extreme distortions (Jacobian determinants) and cortical vs subcortical Dice scores and validation sets. Future work will make these learnable and spatially-varying to support more flexible alignment.
>
> In response to the 'Detailed comments' section, we fixed typos ('regsitration', 'CVSR') and revised the p-value claim to note deep gray matter ($p=0.59$) as the exception.
>
> ## Q3: Additional Baselines
>
> We agree that any final version of this method should be compared against state-of-the-art learning-based frameworks such as VoxelMorph, but at this stage of development, the resolution of our deformation field and label space remain relatively low. Moreover, the choice of volumetric architecture isn't core to the design, so should VoxelMorph outperform PDD, then, in principle, it could be used instead.
>
> The purpose of this paper is instead to validate whether the proposed fully differentiable, joint optimised surface and volumetric alignment framework improves performance over volumetric alignment alone, and this is what we have shown, particularly for challenging cases. As the reviewer acknowledges, this is a challenging and clinically relevant problem for which there is no pre-existing solution, but for clearer reference, we benchmark against FreeSurfer's CVS—the established classical baseline for combined surface-volumetric registration. The results displayed in new Figures 4.2 & 4.3 show that CVS fails for our neonatal cases.
>
> ## Q4: DDR-only Results
>
> As requested, we ran a DDR-only ablation evaluating the surface warp in volumetric space without PDD refinement. Results are shown in Figure 4.1 and Table 1.
>
> ## Q5: Jacobian Analysis
>
> Similarly, we now include detailed Jacobian statistics (mean, std, min, % negative) for CSVR and all baselines, in Table A2.
>
> ## Supplementary responses
>
> ### Geodesic distance-based blending
>
> We thank the reviewer for the suggested use of geodesic distance-based blending when interpolating the surface warp into the volume. Our plan for future work is to adaptively learn the weight coefficients of the interpolation while training, which we hope should achieve the same goal.
>
> ### Warp Inversion
>
> We clarify that no explicit inversion occurs. The target surface is resampled onto moving topology (establishing vertex correspondence), then displacement is computed as $d_i = v_i^{\\mathrm{moving}} - v_i^{\\mathrm{resampled\\\\_target}}$, with vectors showing displacements from target to moving in line with the volumetric standard.
>
> ### 90-Cluster Scheme
>
> This is a valid question that speaks more to the method of Guo et al. (2025), for which it is being addressed in response to a reviewer of that paper. The number represents the best trade-off between cluster size and representational power.
>
> ### Surface Extraction Error Propagation
>
> This is a good point. For cases where surface extraction fails or generates very poor surfaces, downstream alignment will definitely suffer. However, in generally successful cases, even with minor surface extraction errors, significant cortical alignment improvements are observed compared to volumetric-only methods when using surface-based spherical registration. Often, surface extraction failures of this kind stem from segmentation difficulties or poor tissue contrast (due to motion artefacts, partial volume effects, low resolution, etc.)—the same factors that also degrade volumetric registration. Thus, both approaches face similar upstream challenges, but surface-based methods still provide superior cortical correspondence when surfaces can be aligned in the spherical domain. That being said, we agree that visual QC of failing cases, including those that are outperformed by PDD, is clearly of great importance.
>
> ### Narrow evaluation scope
>
> We agree with the reviewer that more metrics of comparison would be beneficial, and we will calculate sulcal depth correlation for the camera-ready version of the paper if given the opportunity. Unfortunately, we had to prioritise validation against CVS and Jacobian analysis due to time constraints, but based on the surface correspondences shown in Figure 4.2, we would expect to also see improvements for sulcal depth metrics.

---

> > ### Comment · Area_Chair_WtS9 · 2026-02-01
> > **Update the final ratings**
> >
> > Please kindly review the authors’ rebuttal and update the final rating by February 1, 2026 (23:59 AoE).

---

### Author Rebuttal · Authors · 2026-01-24

**Rebuttal:**

We thank the reviewers for their positive feedback and constructive criticism. In summary, we have added: (1) CVS baseline comparison, including qualitative image comparisons for T2s, segmentations, and anatomical (WM) surfaces (2) DDR-only ablation, and (3) detailed Jacobian analysis for all methods. We attach the new revised manuscript here.

**Supporting Material:**

/attachment/e268a760f7a04238fbf1a73c6189b1123fc2daa0.pdf

---

### Meta-Review · Area_Chair_WtS9 · 2026-02-09

**Recommendation:** Accept (Oral)
**Confidence:** 5

**Metareview:**

All reviewers recommend an Accept, agreeing that the proposed CSVR framework effectively bridges the gap between surface and volumetric registration for neonatal brain MRI. During the rebuttal, the authors addressed the reviewers' concerns by providing a detailed Jacobian analysis and demonstrating that the method substantially outperforms classical baselines, such as FreeSurfer's CVS, in both accuracy and speed.

---

### Decision · Program_Chairs · 2026-02-13

Accept (Poster)